# A Highly Configurable Packet Sniffer Based on Field-Programmable Gate Arrays for Network Security Applications

Marco Grossi [1,2] , Fabrizio Alfonsi [1,3], Marco Prandini [4] and Alessandro Gabrielli [1,3,*]

1 Department of Physics and Astronomy "Augusto Righi" (DIFA), Alma Mater Studiorum—Università di Bologna, 40126 Bologna, Italy; marco.grossi8@unibo.it (M.G.); fabrizio.alfonsi@bo.infn.it (F.A.)
2 INFN CNAF, 40127 Bologna, Italy
3 INFN Bologna, 40127 Bologna, Italy
4 Department of Computer Science and Engineering, Università di Bologna, 40126 Bologna, Italy; marco.prandini@unibo.it
* Correspondence: alessandro.gabrielli@unibo.it; Tel.: +39-051-209-5052

**Abstract:** Web applications and online business transactions have grown tremendously in recent years. As a result, cyberattacks have become a major threat to the digital services that are essential for our society. To minimize the risks of cyberattacks, many countermeasures are deployed on computing nodes and network devices. One such countermeasure is the firewall, which is designed with two main architectural approaches: software running on standard or embedded computers, or hardware specially designed for the purpose, such as (Application Specific Integrated Circuits) ASICs. Software-based firewalls offer high flexibility and can be easily ported to upgradable hardware, but they cannot handle high data rates. On the other hand, hardware-based firewalls can process data at very high speeds, but are expensive and difficult to update, resulting in a short lifespan. To address these issues, we explored the use of an (Field-Programmable Gate Array) FPGA architecture, which offers low latency and high-throughput characteristics along with easy upgradability, making it a more balanced alternative to other programmable systems, like (Graphics Processor Unit) GPUs or microcontrollers. In this paper, we presented a packet sniffer designed on the FPGA development board KC705 produced by Xilinx, which can analyze Ethernet frames, check the frame fields against a set of user-defined rules, and calculate statistics of the received Ethernet frames over time. The system has a data transfer rate of 1 Gbit/s (with preliminary results of increased data rates to 10 Gbit/s) and has been successfully tested with both ad hoc-generated Ethernet frames and real web traffic by connecting the packet sniffer to the internet.

**Keywords:** network security; packet sniffer; packet classification; FPGA; embedded systems

## 1. Introduction

The Internet of Things (IoT) has interconnected embedded technologies, causing networks to become more complex due to the growth of web applications and business transactions. Consequently, networks have become more susceptible to cyberattacks, which can lead to unauthorized access, denial of service, and theft or alteration of sensitive data [1–3]. Cyberattacks pose a threat in various fields, such as Industry 4.0 [4–6] and healthcare [7–9].

Firewalls and packet sniffers are two common types of network security systems that can be used to prevent cyberattacks and protect computer networks. Firewalls work by monitoring and controlling incoming and outgoing network traffic based on pre-defined rules, while packet sniffers analyze network traffic in real time to detect and isolate potential threats. With the increasing complexity of computer networks and the growing number of cyberattacks, it is becoming more important than ever to have effective network security

measures in place. These devices aim to detect any malicious traffic by either comparing packets with known attack samples (the signature-based approach) or by identifying abnormal traffic that does not match with known legitimate packets (the anomaly-based approach). Our prototype was designed using the signature-based approach, but we also acknowledge the possibility of using other methods to prevent cyberattacks.

The difference between a firewall (or intrusion prevention system, IPS) and a packet sniffer (or intrusion detection system, IDS) is that an IPS possesses, either physically or logically, two ports. When the network data enter one port, these are transparently transferred to the output of the other port if no threats are detected or blocked otherwise [10]. An IDS, on the other hand, works in passive mode, reading all data received at its input port, and sending alarm messages to a remote server when potential threats are detected [11,12]. For home or business networks, where the speeds rarely exceed 1 Gbps, firewalls and packet sniffers can be effectively implemented in software running on a server. In 2017, Nivedita and Kumar proposed an innovative approach for a firewall using a hybrid frame of Netfilter for Linux web servers [13]. In 2016, Nivethan and Papa proposed a new methodology that extended existing Linux-based firewalls to protect the US smart grid, specifically for systems that use the DNP3 protocol. The aim of this approach was to prevent cyberattacks and ensure the security of the smart grid, which is becoming increasingly important in today's interconnected world [14]. In 2022, Tirumala and colleagues conducted a study on the hardware capabilities of Raspberry Pi network interfaces to handle high volumes of incoming traffic for protecting small- and medium-sized enterprises and smart homes [15]. In 2015, Phalguni and Krishna presented a software firewall for the application layer running on an ARM9-based single-board computer based on the Iptables/Netfilter frame in Linux [16]. In 2013, Oluwabukola et al. proposed Psniffer, a packet sniffer software application for network security in Java [17]. In 2008, Phang et al. presented V6SNIFF, an efficient packet sniffer capable of analyzing Ipv6 packets [18]. In 2017, Goyal et al. conducted a comparative study between the two most popular packet sniffing software tools (Tcpdump and Wireshark) [19].

Firewalls and packet sniffers that are based on software running on a standard computer are generally reliable in most situations. However, when the amount of data being transferred and the transfer speed increase beyond a certain threshold, these systems may lose their effectiveness. In such situations, it is preferable to use a hardware implementation since it can guarantee real-time operations and much higher data throughputs. It is common for commercial firewall and packet sniffer products to be based on application-specific integrated circuits (ASICs), which are highly optimized devices designed and manufactured for a specific application. They offer top performances in terms of speed, power consumption, and production cost per unit. However, ASICs are also known for their complex design and high non-recurrent design costs, which make them ideal for high production volumes. Field-programmable gate arrays (FPGAs) are more suitable for products designed for small production volumes and research projects. These semiconductor devices feature quick design steps and negligible non-recurrent engineering costs. In 2019, Niemiec et al. conducted a survey addressing the open research challenges that need to be tackled for the adoption of FPGAs in accelerating virtualized network functions [20]. In 2011, Wicaksana and Sasongko presented a prototype of a hardware stateless firewall designed using Cyclone II FPGA working at 91 MHz [21]. According to these authors, the implementation only included packet classification, and due to the absence of an efficient FIFO buffer, it hinders high-speed data transfer. In 2017, Lin et al. presented an Ethernet firewall based on a FPGA that achieves a data throughput of 950 Mbit/s. The FPGA can be interfaced to ARM devices to realize a management server [22]. In 2012, Prajapati and Khare proposed a framework for a firewall hardware on a FPGA designed in Verilog that can handle IPv4 and IPv6 network data [23]. In 2020, the same authors presented a reconfigurable firewall based on a Xilinx Virtex-6 FPGA. This firewall achieves a throughput of 142 Gbit/s at a clock rate of 442.6 MHz for a minimum packet size of 40 bytes [24]. Mohammed and Ueno proposed a FPGA-based firewall in 2018, which was based on a Xilinx Kintex-7 XC7K325T

device. The proposed firewall exploits content addressable memory to achieve a much better performance than a Linux firewall based on Iptables [25]. In 2021, Hilgurt presented a brief overview of various approaches, methods, and techniques used for designing a FPGA-based IDS [26]. In 2022, Stój et al. introduced an Ethernet packet sniffer based on a Xilinx Kintex-7 FPGA platform, which was designed for network intrusion detection in Industry 4.0 [27]. In 2013, Pal et al. proposed E-Sniff, a small special-purpose embedded system for capturing and logging network data based on a Cyclone II FPGA [28]. In 2005, Song and Lockwood published a paper on an efficient packet classification system for network intrusion detection using a FPGA that achieved a data rate of 2.5 Gbit/s [29]. Faria et al.'s FPGA-based Ethernet sniffer for real-time networks, which they proposed in 2009, can be interfaced via USB to a host computer for the generation of graphics and statistical data [30]. In 2016, Fiessler and colleagues proposed a hybrid packet classification approach called HyPaFilter. This approach handles simple operations in hardware designed with a FPGA, while complex operations are dealt with using a Linux-based software firewall [31]. In 2010, Ezzati et al. proposed a packet classification engine based on artificial neural networks that achieves a 97% accuracy in the classification of TCP/IP packets [32]. The authors of the paper proposed a method that they deemed suitable for FPGA implementation. They carried out Matlab simulations and synthesized the VHDL code, but they did not develop an operative system.

In this work, an Ethernet packet sniffer based on a FPGA was proposed. This system was intended for applications in data protection for universities and research institutes. As discussed by Ulven and Wangen in 2021, data breaches and cyberattacks represent a severe problem in higher education institutions and universities [33]. Since we estimate a low number of produced devices, this system was designed on a FPGA, which allows for quick development and negligible non-recurrent engineering costs. Moreover, this system has been designed for easy reconfigurability, where the rules to discriminate between safe and potentially dangerous data can be decided by the user and uploaded via serial communication using a PC.

## 2. Paper Structure

This paper describes a hardware-based packet sniffer that was built using Verilog and a Xilinx KC705 development board (AMD-Xilinx, USA). This system is highly customizable and has been tested for a data transfer rate of 1 Gbit/s, with some preliminary results obtained for a data transfer rate of 10 Gbit/s. Users can configure rules to distinguish between safe and potentially dangerous Ethernet frames using software designed in LabVIEW (ver. 2023 Q3). This system also provides statistics on the network traffic being analyzed. This paper is structured as follows: Section 3 presents the format of an Ethernet frame and the most important fields for identifying potential threats. Section 4 describes the packet sniffer, including its signal characteristics and simulation behavior. Section 5 presents the test results for the designed packet sniffer, compared with results obtained using Wireshark, a popular packet-sniffing software. Section 6 presents the preliminary results for extending the project to higher data rates (10 Gbit/s). In Section 7, the characteristics of the proposed packet sniffer are compared with similar systems developed on a FPGA from the literature. Finally, Section 8 presents the conclusions.

## 3. Ethernet Frame Format

Ethernet is a widely used networking technology that finds its application in local area networks (LANs), metropolitan area networks (MANs), wide area networks (WANs), and other fields, such as industry, avionics, telecommunication, and multimedia. The Ethernet technology was introduced in 1980, and the first standardization was conducted in 1983 by IEEE 802.3 [34]. The data transfer rate of Ethernet has evolved from the initial 2.94 Mbps up to 100 Gbps [35]. In Ethernet networking, the data are described according to different levels of abstraction in the OSI model [36]. At level 2 (data link layer) of the OSI model, the data are described in the form of frames. The format of an Ethernet II frame is displayed in

Figure 1. It always starts with a preamble and a start frame delimiter (SFD), and it ends
with a frame checksum (FCS), a four-byte CRC that detects any corrupted data in the frame.
The header of the Ethernet frame consists of the following fields: a destination and source
media access control (MAC) address, a VLAN tag (an optional field), and the protocol type
of level 3 (network layer), while the payload represents the data of the network layer. The
packet formats of two of the most important protocols of the network layer, ARP, and IP,
are presented in Figures 2 and 3, respectively. The IP header also includes the protocol of
the level 4 layer (transport layer), whose data are present in the IP payload. Three different
protocols of the transport layer were considered, namely TCP, UDP, and ICMP, and their
packet formats are presented in Figures 4–6, respectively.

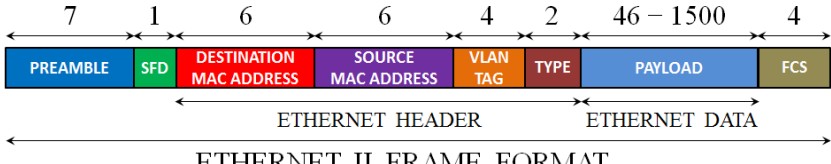

**Figure 1.** The format of an Ethernet II frame. The length of each field of the frame is reported as the
number of bytes.

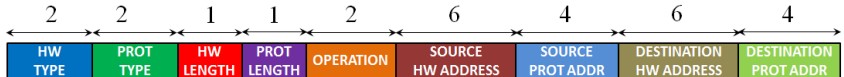

**Figure 2.** The format of an ARP packet. The length of each field of the frame is reported as the
number of bytes.

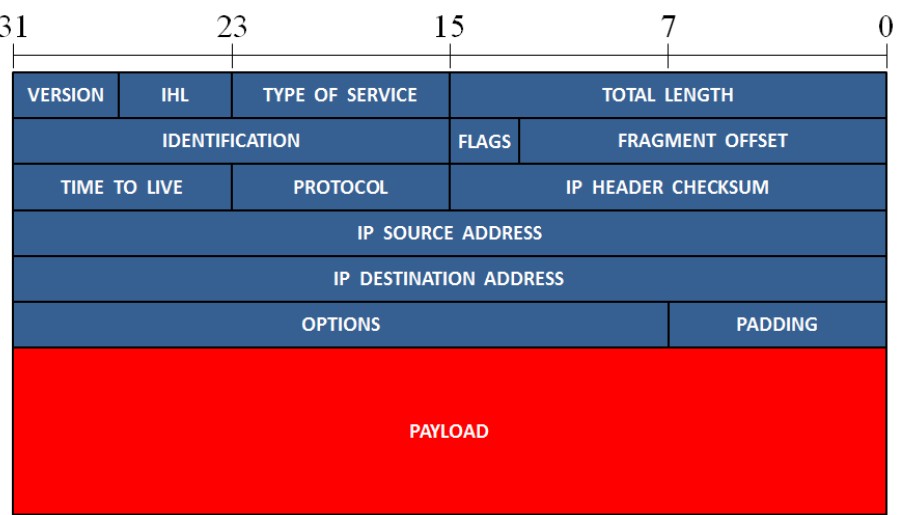

**Figure 3.** The format of an IP packet. The different fields of this packet are presented in rows of
32 bits.

The fields that are most important from the perspective of cybersecurity for the TCP-IP
and UDP-IP packets are the IP source and destination addresses, along with the source
and destination ports. The IP address, along with the MAC address, is used to identify
a device within a network. Moreover, while the IP address is used as a global address,
the MAC address is utilized as a local address. When data are directed on the internet,
the IP address is used, and when the local area network (LAN) has been reached, the IP
address is translated to the MAC address, and the data is delivered to the intended device.
Similarly, the source and destination port numbers are utilized to define the application
or service involved, so that the operating system can deliver the packet to the appropriate
process. The importance of these parameters for cybersecurity is two-fold. Being able to
read them enables a potential attacker to infer who is communicating with whom (in terms

of network hosts) and what kind of dialogue is happening, even if the contents (payload) are encrypted. It is not possible to obscure IPs and ports without modified protocols, such as IPSec [37], or by encapsulation performed by specific applications, such as virtual private networks (VPNs). Obscuring IPs and ports using these techniques may interfere with the functionality of a firewall or a packet sniffer. Therefore, encrypted IP and port numbers were not considered. An attacker that can alter these parameters can hide their own identity by spoofing the source address, usually with the goal of bypassing firewall rules that would block packets bearing their real address.

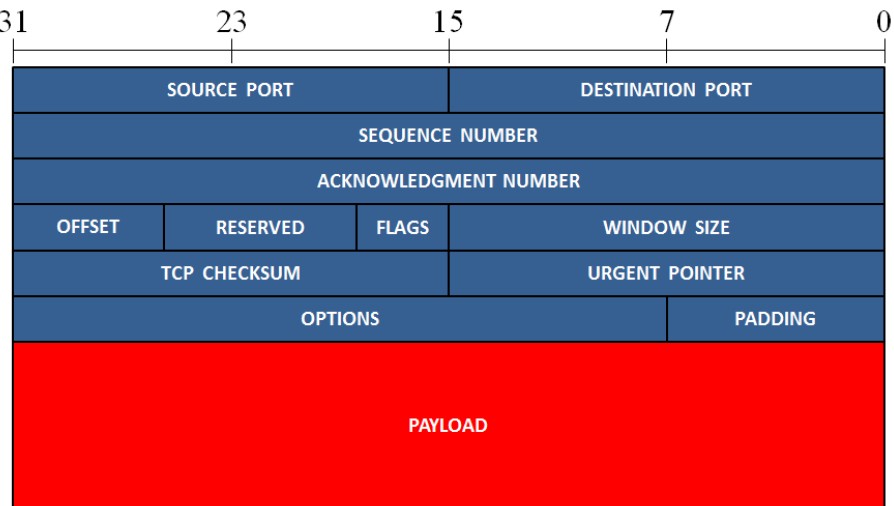

**Figure 4.** The format of a TCP packet. The different fields of this packet are presented in rows of 32 bits.

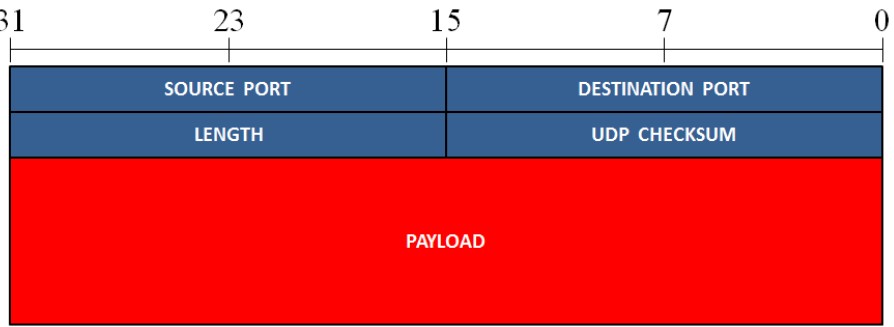

**Figure 5.** The format of an UDP packet. The different fields of this packet are presented in rows of 32 bits.

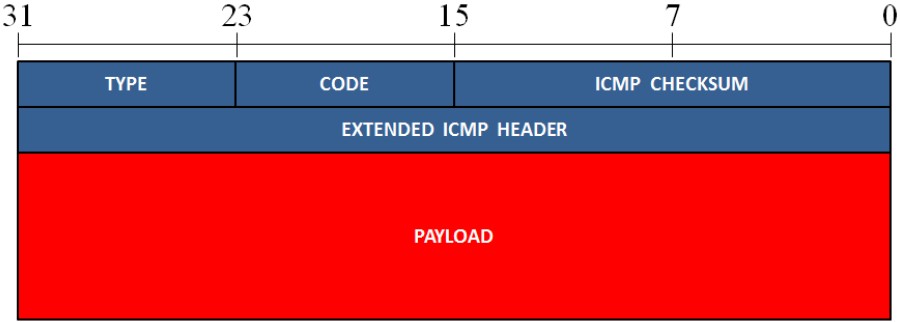

**Figure 6.** The format of an ICMP packet. The different fields of this packet are presented in rows of 32 bits.

## 4. Experimental Design

The experimental setup to validate the developed packet sniffer consists of two parts. The first part is a packet generator that generates Ethernet frames with variable data rates, which can be selected by the user. The second part is the packet sniffer that analyzes the Ethernet traffic. The analysis includes Ethernet data statistics based on the fields of the analyzed Ethernet frames. To ensure the packet sniffer's accuracy, using a packet generator instead of a PC with software to generate Ethernet frames is important, which may produce uncertain results due to uncontrolled data traffic (a PC with a network card generates control frames in addition to the target Ethernet frames). The experimental setup included a packet generator that produces Ethernet frames, with selectable variable data rates, and a packet sniffer that analyses Ethernet traffic and generates data statistics based on frame fields.

The FPGA development board KC705 produced by Xilinx [38] was used to design both the packet generator and the packet sniffer. The development board was equipped with a Kintex-7 XC7K325T-2FFG900C FPGA, 1 GB of DDR3 RAM, 128 MB of flash memory, a 200 MHz LVDS oscillator, PCI Express connectivity, USB ports, and a 10/100/1000 tri-speed Ethernet PHY (Marvell M88E1111-BAB1C000). It also has push buttons, switches, and LEDs for user interaction. The tri-speed Ethernet PHY chip, which is integrated on the board, manages the physical layer of the Ethernet protocol, while the tri-mode Ethernet media access controller (TEMAC), an intellectual property module provided by Xilinx, manages the data link layer [39]. The packet generator and the packet sniffer can generate and analyze Ethernet traffic at a rate of 1 Gbit/s. The Ethernet frame analysis included the ARP, IPv4, UDP, TCP, and ICMP protocols, with a maximum frame size of 1518 bytes for normal frames and 1522 bytes for VLAN frames. For more details on the implementation of the packet generator and the packet sniffer, refer to Sections 4.1 and 4.2 of this article.

### 4.1. The Packet Generator

To assess the functionality of the packet sniffer, a KC705 development board was used to generate Ethernet frames. Figure 7 illustrates the main functional blocks that make up the packet generator, including all interconnections among the blocks. The generator utilizes a 200 MHz differential clock available on the board (CLK_IN_N and CLK_IN_P) and the serial UART receiving port (UART_RX). The 200 MHz differential clock was used to generate two clock signals: UART_CLK (10 MHz) and MAC_TX_CLK (125 MHz), which are used to provide the clock signal for various modules. The UART_RX module reads the serial input UART_RX and generates an eight-bit output byte (UART_RX_DATA) that is sampled when the signal UART_RX_DV is active. The process involves the UART_RX_CONTROL module receiving eight-bit data from UART and storing it in the 16 kbyte-distributed RAM of the MEMORY module. After the data are loaded, the MAC-TX CONTROL module fetches the frame data from memory and passes it on to the TRI-MODE_ETHERNET_MAC module, which generates the Ethernet frames. Figure 8 displays the waveforms of the signals for the case of frame data loading in the memory. This example includes a set of two Ethernet frames, each with a length of 60 bytes, encapsulated with UDP and TCP data. This operation starts when the UART_RX_CONTROL module receives the control byte 0x14, followed by two bytes indicating the number of frames loaded (0x0002), the total number of bytes that follow (0x007c), and, for each frame, the data preceded by two bytes indicating the frame length (0x003c). The data are then stored in a 16kbyte-distributed RAM, with synchronous write and asynchronous read in the MEMORY module. Figure 9 shows the waveform signals at the end of the frame dataset loading.

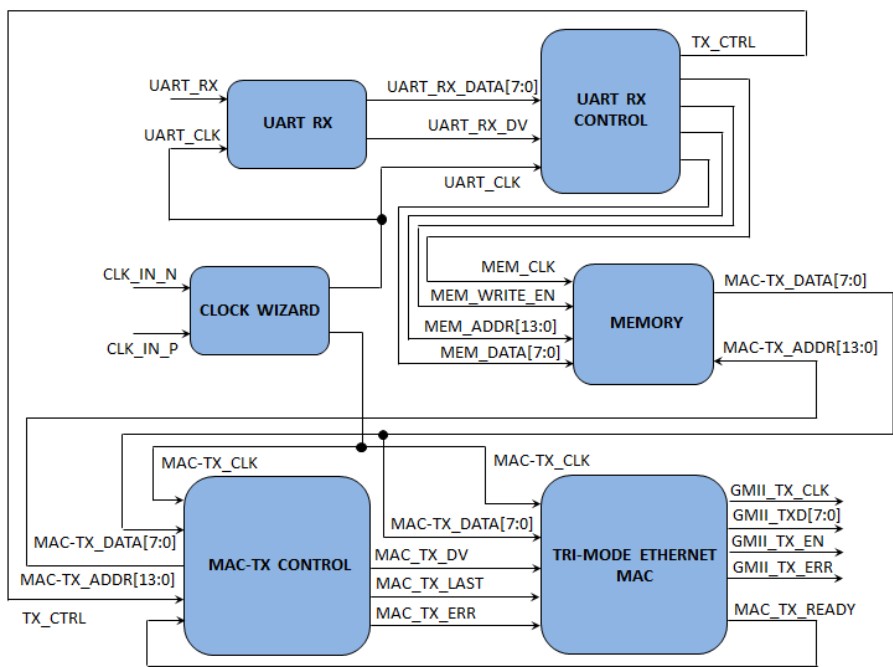

**Figure 7.** A schematic of the packet generator and its main functional blocks.

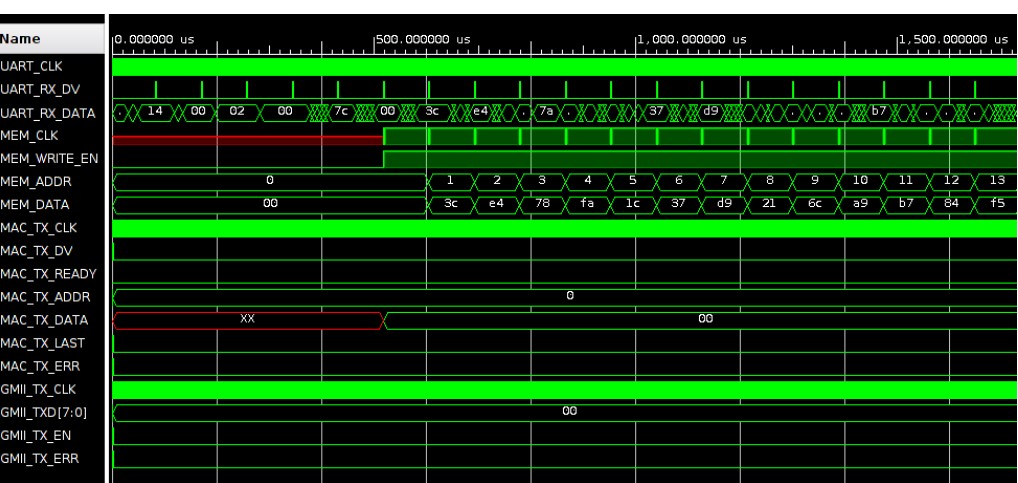

**Figure 8.** Signal waveforms in the case of data loading in memory using UART (start of the operation).

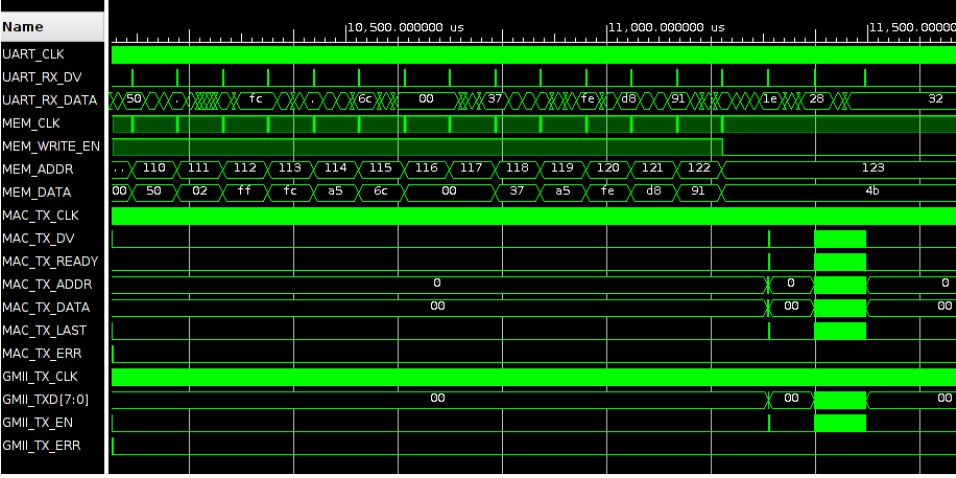

**Figure 9.** Signal waveforms in the case of data loading in memory using UART (end of the operation).

After loading the frame dataset into memory, various operations can be executed by sending the appropriate control byte to the UART_RX_CONTROL module. For instance, sending control byte 0x1e (decimal value 30) initiates a single write operation, causing the Ethernet port to send out the frames in the dataset. Alternatively, control byte 0x28 (decimal value 40) initiates a continuous write operation, where the frames in the dataset are sent out on the Ethernet port in a looping mode. To stop this continuous write operation, a control byte 0x32 (decimal value 50) is used. Additionally, this system can be programmed to include a specified delay between the end of one frame and the start of the next, allowing for a controlled data rate of Ethernet transmission. As displayed in Figure 9, the control byte 0x1e is sent after the end of frame loading (last byte 0x4b at address 123) to output a single set of two Ethernet frames. Then, the control byte 0x28 is sent to start a looping data transfer. The data transfer is finally terminated by sending the control byte 0x32.

Figure 10 shows the waveform signals in the case of a single write operation, where the two Ethernet frames in the dataset are read from memory via the MAC-TX_CONTROL module. Then, the data are sent to the input of the TEMAC module, which generates the output data for the Ethernet PHY chip on the KC705 board. Figures 11 and 12 illustrate the signal waveforms of the first frame's transmission. The valid data signal (MAC_TX_DV) was enabled during the transmit operation, and data read from memory were sampled when the MAC_RX_READY signal was active. Initially, the frame length in bytes was read from memory (0x003c, 60 bytes); then, the MAC_TX_DV signal was enabled, and the frame data were transmitted. During the transmission of the last byte of the frame, the MAC_TX_LAST signal was enabled; then, the MAC_TX_DV signal was disabled, and, after a delay, the second frame was transmitted. At the output of the TEMAC module, the frame data (GMII_TXD) were sent to the Ethernet PHY chip preceded by the frame preamble (0x55555555555555d5) and followed by the frame checksum (0x8a68866a). During transmission, the GMII_TX_EN signal was enabled.

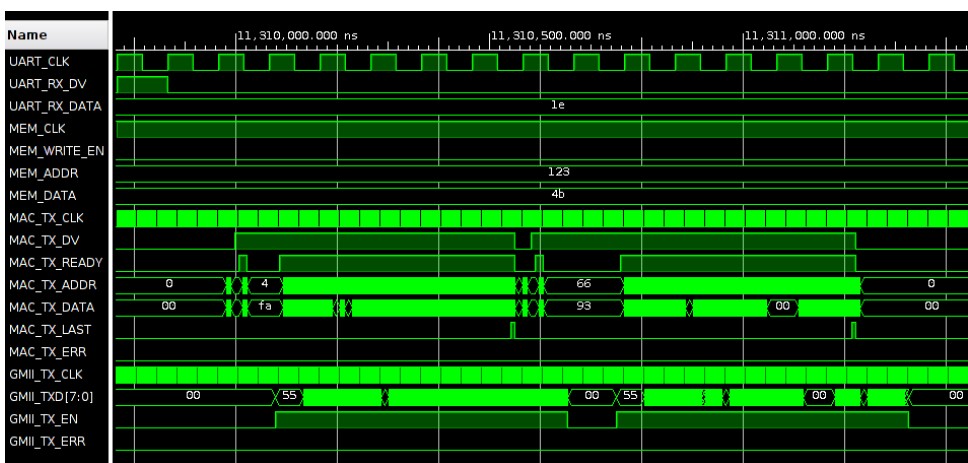

**Figure 10.** Signal waveforms in the case of the generation of two Ethernet frames in sequence.

Programs were written in LabVIEW to control the packet generator. The tests were carried out by connecting the packet generator's Ethernet port to an Ethernet port on the PC and generating several sets of Ethernet frames. The network traffic on the PC was analyzed using Wireshark to verify the correct functioning of the packet generator.

### 4.2. The Packet Sniffer

Figure 13 illustrates the core functional blocks comprising the packet sniffer, including visualizing all the interconnections among these blocks.

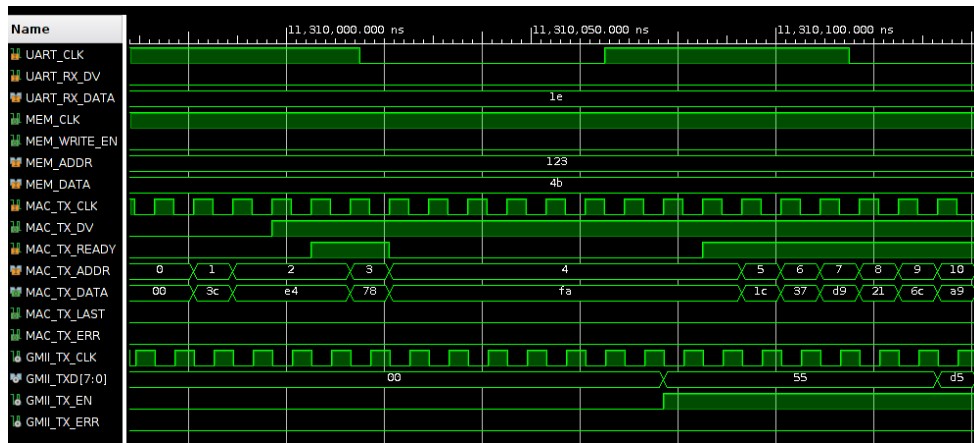

**Figure 11.** Signal waveforms in the case of the generation of an Ethernet frame (start of the operation).

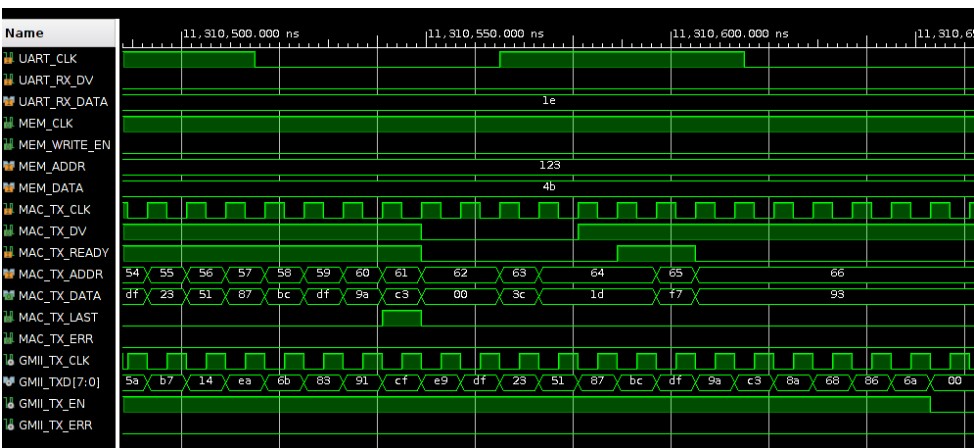

**Figure 12.** Signal waveforms in the case of the generation of an Ethernet frame (end of the operation).

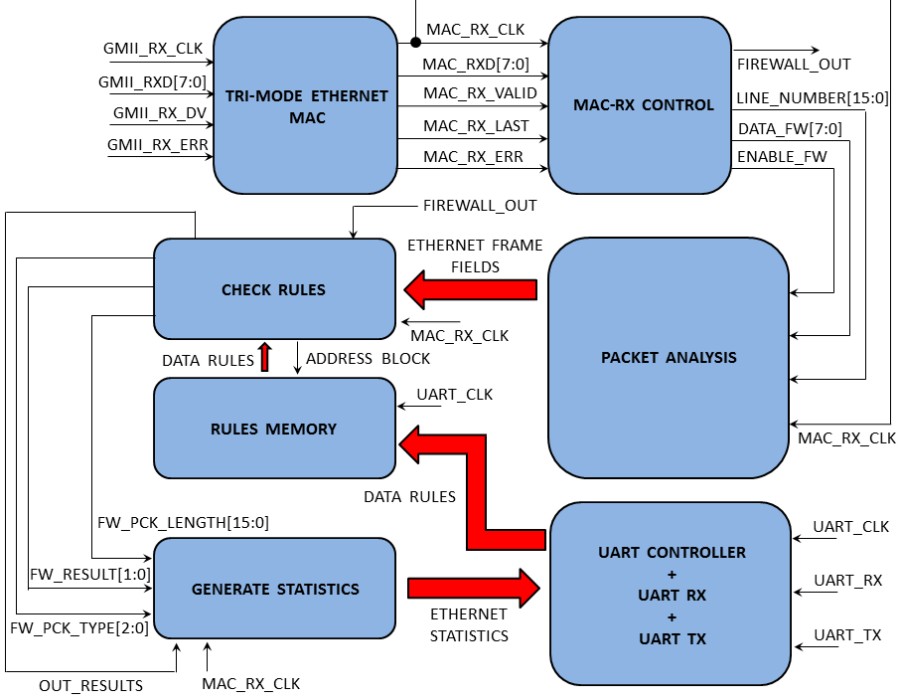

**Figure 13.** A schematic of the packet sniffer and its main functional blocks.

Two clocks were derived from the 200 MHz differential clock on the KC705 board: MAC_RX_CLK (125 MHz), used for packet reception and analysis synchronization, and UART_CLK (10 MHz), used for data transmission/reception between the packet sniffer and the PC via UART at a 9600 baud rate.

Ethernet frames were received by the Marvell M88E1111-BAB1C000 Ethernet PHY chip on the KC705 board, transmitted in parallel (1 byte) to the TEMAC IP produced by Xilinx, which removes the frame preamble and validates the frame checking sequence (FCS). The MAC_RX_CONTROL module then forwards the data to the PACKET_ANALYSIS module for field extraction (as discussed in Section 3).

Figure 14 displays signal waveforms for the continuous reception of four different Ethernet frames (UDP, TCP, ARP, and ICMP).

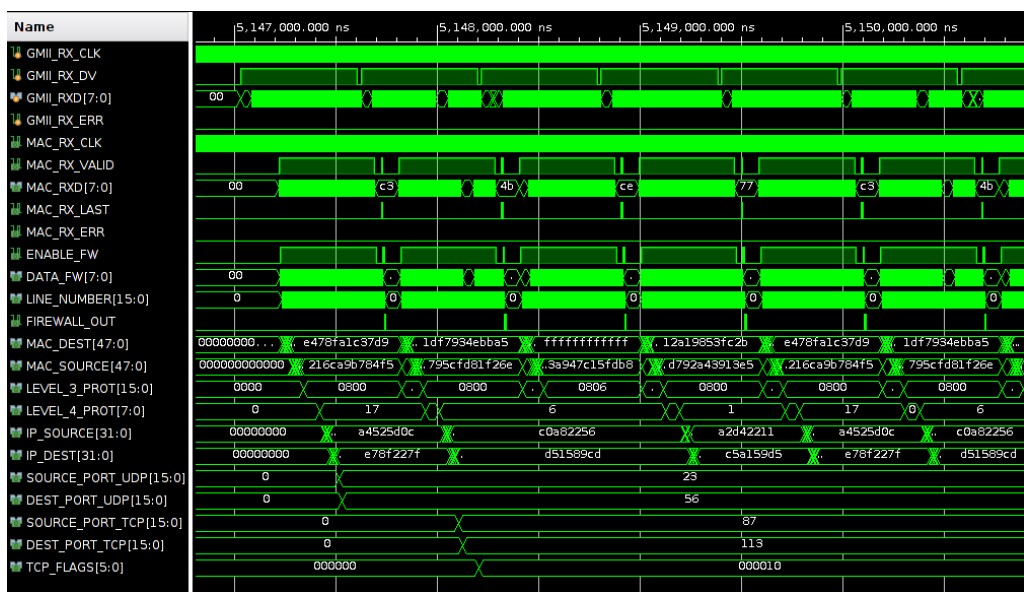

**Figure 14.** Signal waveforms for the reception and analysis of Ethernet frames.

When the packet analysis was running, the ENABLE_FW signal was activated, and upon completion, a pulse was generated on the FIREWALL_OUT signal. This analysis extracted various frame parameters, including MAC source and destination addresses (MAC_SOURCE and MAC_DEST), layer 3 protocol (LEV_3_PROT), and for LEV_3_PROT = 0x0800 (IP packets), layer 4 protocol (LEVEL_4_PROT), IP source and destination addresses (IP_SOURCE and IP_DEST), source and destination ports for the UDP and TCP packets, and flag information for the TCP packets.

After frame analysis and the pulsed FIREWALL_OUT signal, the CHECK_RULES module assesses if the packet matches the firewall rules. These rules are defined based on IP packet characteristics, such as source and destination address ranges, layer 4 protocol, and source/destination port ranges, are stored in the RULES_MEMORY module, and loaded from a PC via UART. The CHECK_RULES module outputs packet type (FW_PCK_TYPE), packet length (FW_PCK_LENGTH), and analysis results (FW_RESULT—zero for packet errors, one for rule violations, and three for allowed packets).

Upon completing the rules check, the OUT_RESULTS signal is pulsed, and the GEN-ERATE_STATISTICS module calculates packet statistics (type, length, allowed/rejected status, total data transferred, etc.), which can be transmitted to the PC via UART.

Figure 15 displays signal waveforms for frame analysis and statistics generation, featuring four continuously received Ethernet frames (UDP, TCP, ARP, and ICMP). In this case, the packet sniffer rules were defined to only allow the IP source address of the TCP packet.

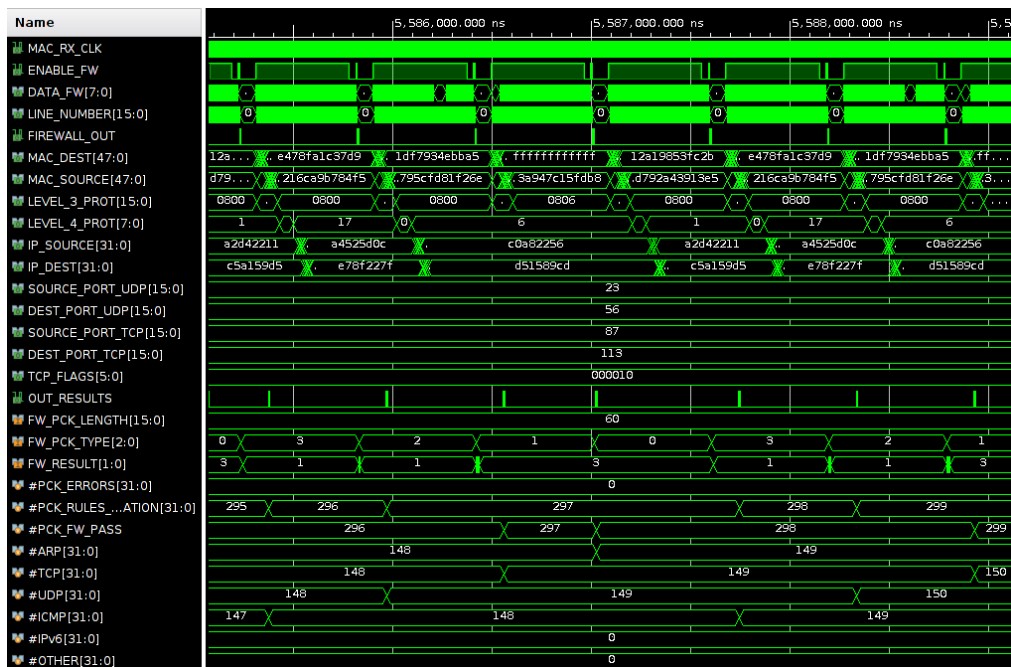

**Figure 15.** Signal waveforms for the frame analysis and statistics generation of Ethernet frames.

As shown in Figure 15, among the four generated packet types (UDP, TCP, ARP, and ICMP), TCP (FW_PCK_TYPE = 1) and ARP (FW_PCK_TYPE = 0) are classified as 'allowed' (FW_RESULT = 3). Specifically, TCP packets were allowed, as their IP source addresses comply with the firewall rules, and ARP packets were allowed as they are not IP packets (LEV_3_PROT = 0x0806).

Conversely, IP packets (LEV_3_PROT = 0x0800), with Level 4 protocols UDP (LEV_4_PROT = 17) and ICMP (LEV_4_PROT = 1), were marked as 'rejected', since their IP source addresses do not match the firewall rules.

## 5. Results

The packet sniffer, as discussed in Section 4.2, underwent performance evaluation. Initially, testing involved the generation of Ethernet frames using the packet generator outlined in Section 4.1, with the results detailed in Section 5.1. Subsequently, the packet sniffer was connected to the internet to assess its reliability and performance under real Ethernet traffic, and the outcomes are presented in Section 5.2.

### 5.1. Tests with the Packet Generator

The packet sniffer was tested by generating Ethernet frames with the packet generator. The measurement setup is depicted in Figure 16.

Considering that the packet sniffer reads a single byte in a clock cycle of duration 8 ns ($T_{\mathrm{CLK}}$), corresponding to a 125 MHz frequency, the number of clock cycles $N_{\mathrm{CLK}}$ needed to analyze an Ethernet frame of byte length $N_{\mathrm{ETH}}$ is given by:

$$N_{\mathrm{CLK}} = N_{\mathrm{PREAMBLE}} + N_{\mathrm{SFD}} + N_{\mathrm{ETH}} + N_{\mathrm{FCS}} + N_{\mathrm{MAC}} + N_{\mathrm{DELAY}} \tag{1}$$

where $N_{\mathrm{PREAMBLE}}$ is the number of clock cycles needed to read the Ethernet frame preamble (7), $N_{\mathrm{SFD}}$ is the number of clock cycles needed to read the Ethernet start frame delimiter (1), $N_{\mathrm{FCS}}$ is the number of clock cycles needed to read the Ethernet frame checksum (4), $N_{\mathrm{MAC}}$ is the number of clock cycles of delay introduced via the TEMAC module, and $N_{\mathrm{DELAY}}$ is the number of clock cycles of delay set in the packet's generator using the UART command. Consequently:

$$N_{\mathrm{CLK}} = N_{\mathrm{ETH}} + N_{\mathrm{MAC}} + N_{\mathrm{DELAY}} + 12 \tag{2}$$

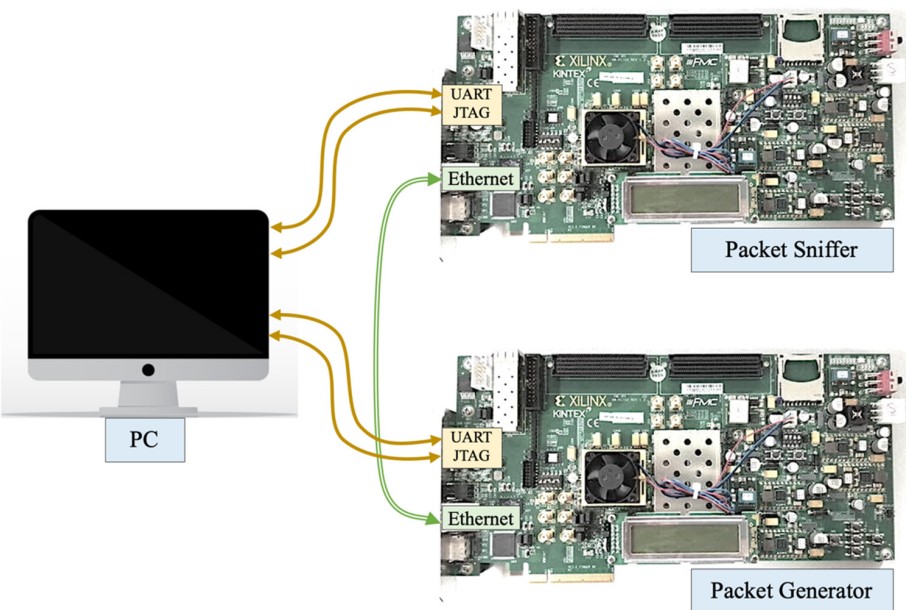

**Figure 16.** Experimental setup, where the Ethernet cables of the packet generator and the packet sniffer are connected.

Thus, the data rate (DR), expressed in bit/s, can be calculated as:

$$\text{DR} = \frac{8 \times N_{\text{ETH}}}{N_{\text{CLK}} \times T_{\text{CLK}}} = 10^9 \frac{N_{\text{ETH}}}{N_{\text{ETH}} + N_{\text{MAC}} + N_{\text{DELAY}} + 12} \tag{3}$$

From Equation (3), it is evident that, even for $N_{\text{DELAY}} = 0$, the data rate cannot reach the maximum value of 1 Gbit/s. However, it can be asymptotically approached as $N_{\text{ETH}} \to +\infty$. We conducted tests using the packet generator in continuous mode, assessing various Ethernet frames with different byte lengths and protocols: ARP (42 bytes), TCP (54 bytes), TCP (74 bytes), UDP (112 bytes), UDP (256 bytes), TCP (506 bytes), TCP (983 bytes), and TCP (1414 bytes). Each Ethernet frame was subjected to different inter-frame delays, including 0, 10, 50, 100, 500, 1000, 5000, 10,000, and 50,000 clock cycles. The results of these tests are presented in Figure 17.

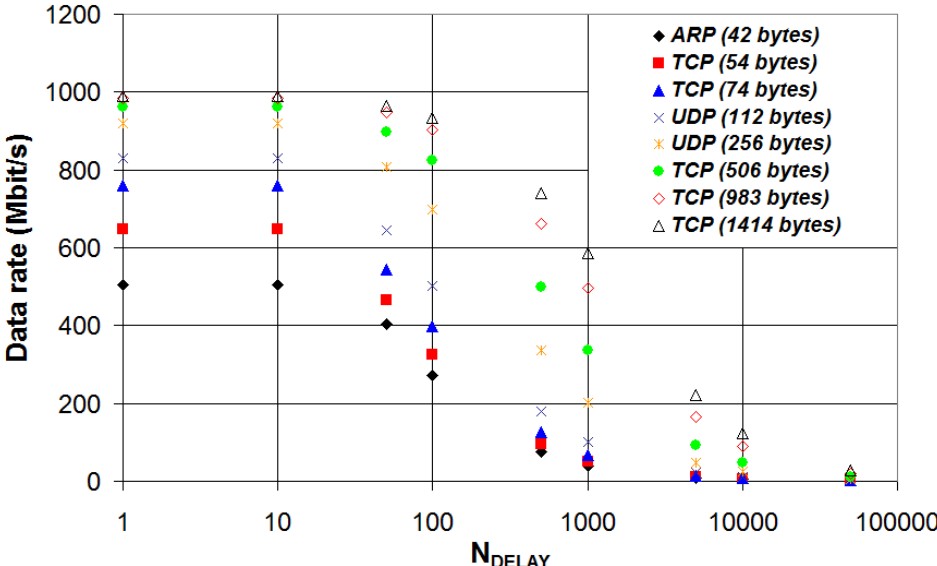

**Figure 17.** Data rate as a function of inter-frame delay for Ethernet frames of different byte lengths and different protocols.

As anticipated, the data rate declined with increasing inter-frame delay. The data, depicted in Figure 17, were fitted to Equation (3), utilizing CurveExpert Professional v2.4.0 (Hyams Development). The results exhibit a strong correlation, with coefficients of determination ($R^2$) ranging from 0.989 to 0.999.

The maximum data rate, $DR_{MAX}$, is achieved when no inter-frame delay is present ($N_{DELAY} = 0$), and it can be expressed as:

$$DR_{MAX} = 10^9 \frac{N_{ETH}}{N_{ETH} + N_{MAC} + 12} \tag{4}$$

In Figure 18, the maximum data rate is plotted vs the frame length in bytes ($N_{ETH}$). These measured data have been fitted to Equation (4), and this resulted in a high correlation ($R^2 > 0.98$) and an estimated value of $N_{MAC}$ of about 16 clock cycles.

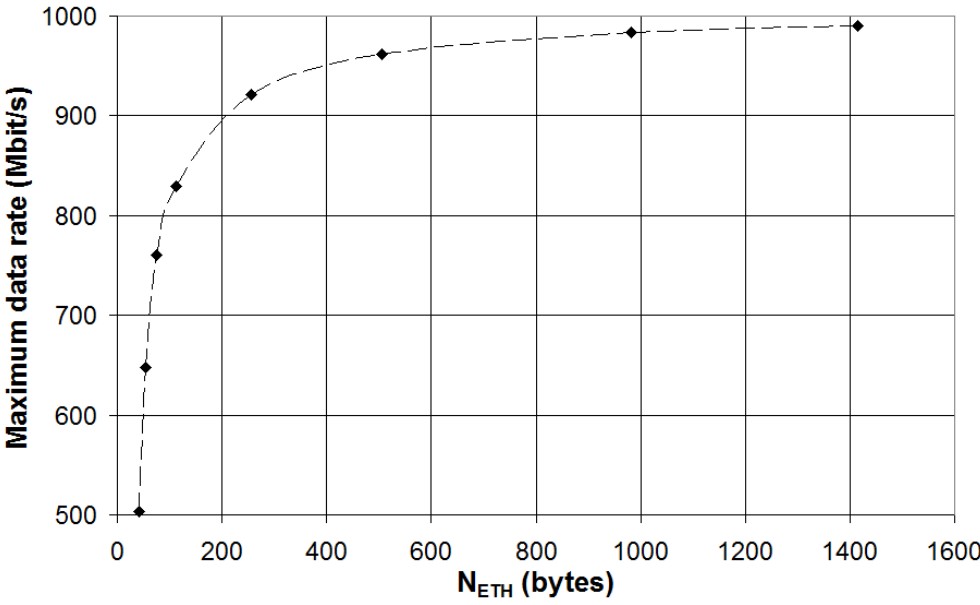

**Figure 18.** Maximum data rate as a function of the Ethernet frame length ($N_{ETH}$).

Next, we evaluated and compared the performance of our proposed packet sniffer against a PC running Wireshark. Figure 19 illustrates the experimental setup, which involved the use of an Extreme Networks Summit X450e-48p switch (Extreme Networks) to provide identical Ethernet traffic to both the packet sniffer and the PC running Wireshark. The PC's Ethernet port was linked to switch port two, while the traffic from port two was mirrored to port ten, where the Ethernet port of the packet sniffer was connected. To accommodate this experimental setup, we implemented the optional virtual local area network (VLAN) field in the Ethernet frame within the packet sniffer. This adjustment was necessary, as the traffic mirrored to port ten carried the VLAN tag. The packet generator was linked to port six of the switch, responsible for generating Ethernet frames addressed to the PC. These Ethernet frames were continually transmitted, with varying inter-frame delay values. The results of this test are illustrated in Figure 20, depicting frame rates over time for both the packet sniffer and the PC running Wireshark. The plotted data were found to align, as expected, but there was a notable delay of approximately 8–11 s in the PC running Wireshark's response compared to the packet sniffer. Furthermore, when the frame rates exceeded 48,000 frames per second, the PC running Wireshark began losing data and eventually crashed at frame rates exceeding 100,000 frames per second. This demonstrates the packet sniffer's ability to accurately monitor Ethernet traffic with data rates of up to 1 Gbit/s, while a PC running Wireshark encounters issues when data rates are too high.

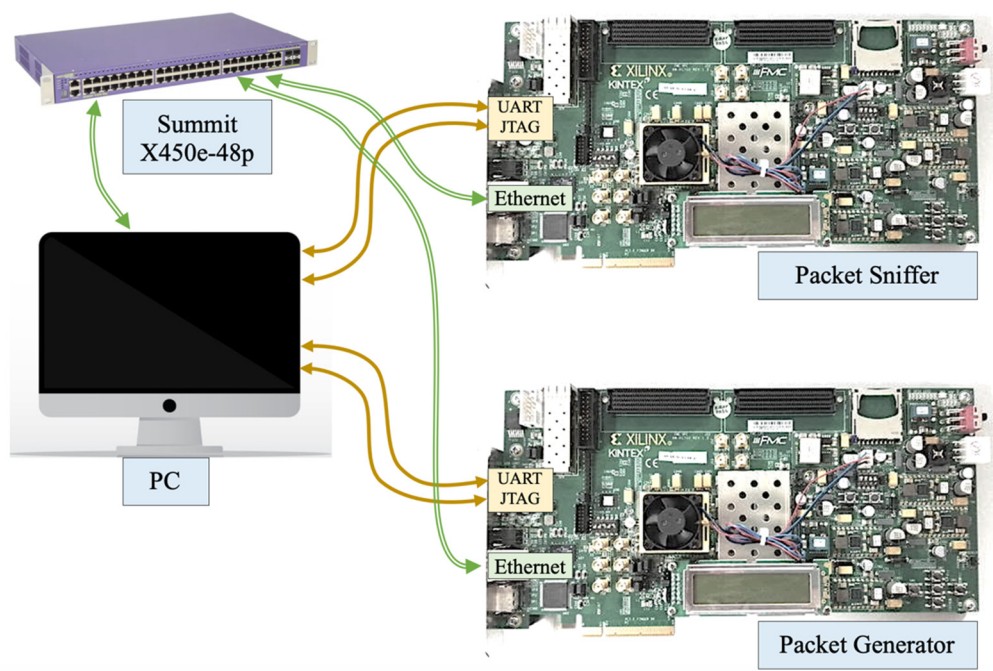

**Figure 19.** Experimental setup, where the PC Ethernet traffic is mirrored on the packet sniffer and the packet generator is used to generate the traffic.

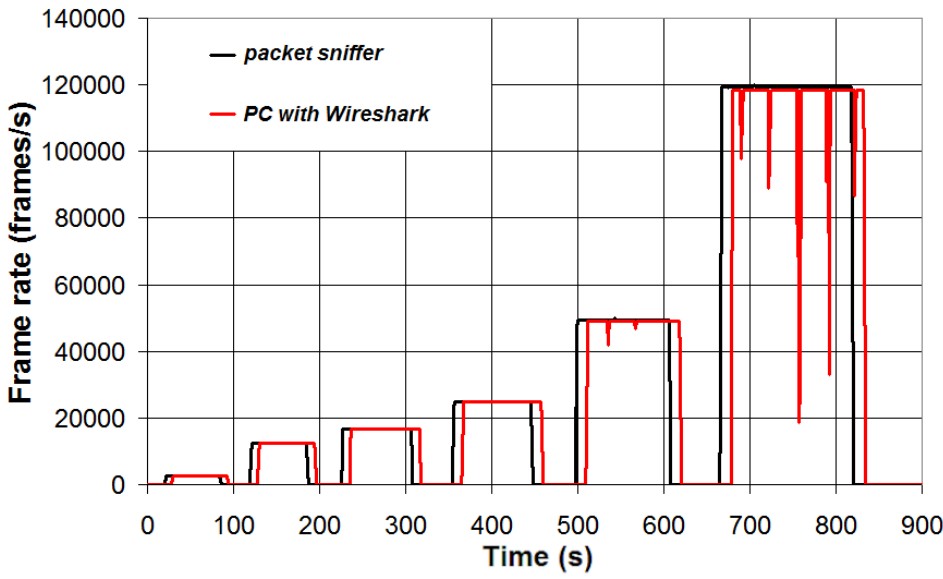

**Figure 20.** Measured frame rate vs time in the case of the proposed packet sniffer and a PC running Wireshark.

### 5.2. Tests with Real Ethernet Traffic

After successfully testing the designed packet sniffer with controlled Ethernet traffic generated via the packet generator, the system was further evaluated with real-world Ethernet traffic. This real-world test was conducted using the experimental setup depicted in Figure 21.

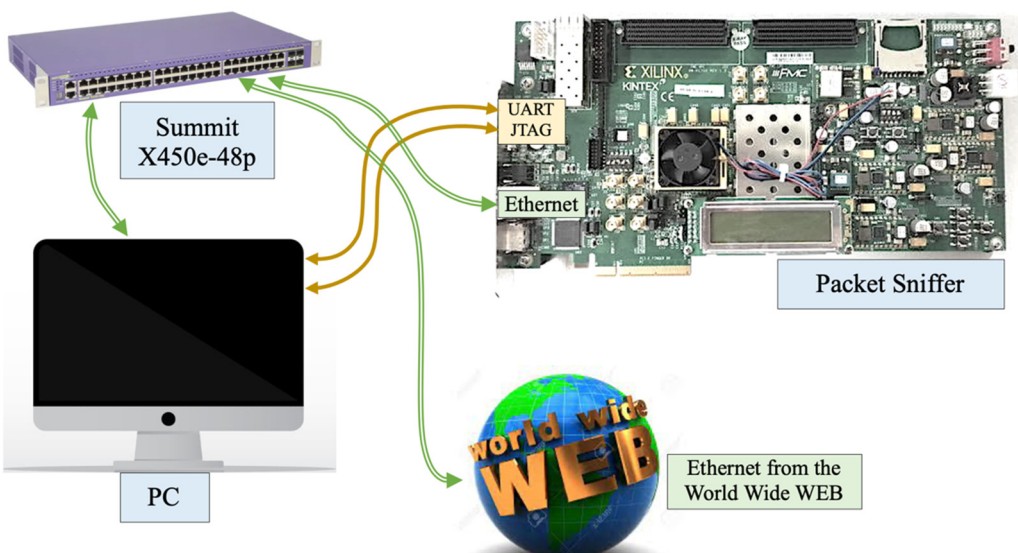

**Figure 21.** The experimental setup, where the PC Ethernet traffic is mirrored on the packet sniffer and real Ethernet traffic is generated via the PC.

In this setup, an Extreme Networks Summit X450e-48p switch was deployed to provide identical internet traffic to both the packet sniffer and the PC running Wireshark. The PC's Ethernet port was connected to port two of this switch, and the traffic on port two was mirrored to port ten, where the Ethernet port of the packet sniffer was connected. To enable internet access for the PC, a router was connected to port six of this switch.

In the initial test, we emulated an ICMP flood attack targeted at the PC connected to port two of this switch. This was achieved using the UDP Flooder software on another PC within the same local area network. Figure 22 displays five distinct attacks, each generated for varying durations. The results include data collected via both the packet sniffer and the PC running Wireshark.

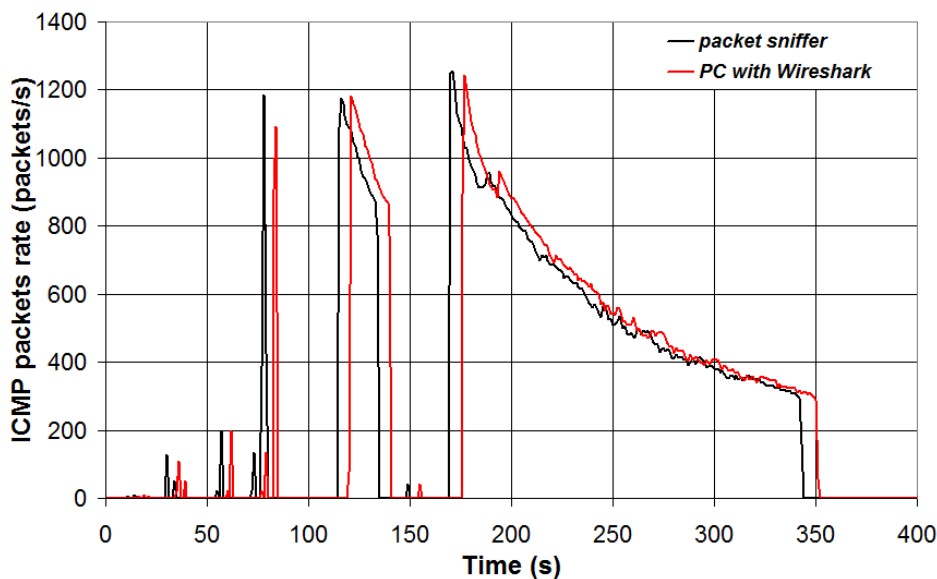

**Figure 22.** ICMP packets rate under different ICMP flood attacks detected with the packet sniffer and the PC running Wireshark.

To assess the data correlation between the packet sniffer and Wireshark, we conducted a cross-correlation analysis utilizing the Real Statistics add-on for Excel. This analysis

revealed a delay of approximately 6 s in the Wireshark data, with a maximum correlation factor estimated at 0.99.

As a second test, we generated UDP Ethernet traffic by streaming content from an online video service. We viewed a 2 min YouTube video on the PC at various video resolutions (at 1080 p, 720 p, and 480 p) and monitored the corresponding UDP traffic using both the packet sniffer and Wireshark. The number of UDP packets detected using the packet sniffer in 1 s is depicted in Figure 23 for the 1080 p video resolution, Figure 24 for the 720 p resolution, and Figure 25 for the 480 p resolution.

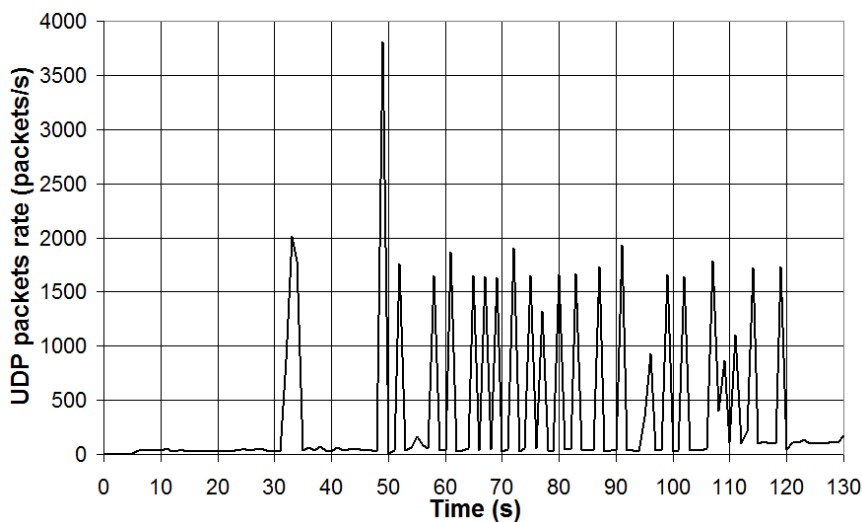

**Figure 23.** UDP packets' rate in the case of the video resolution 1080 p.

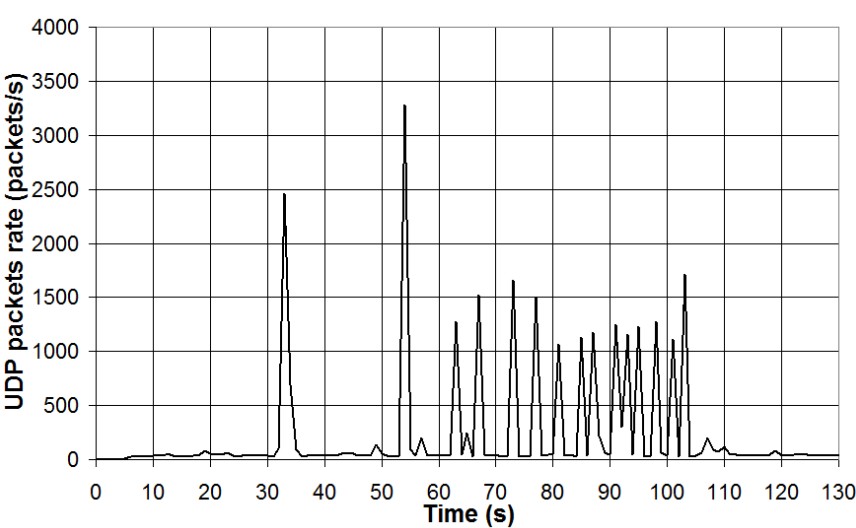

**Figure 24.** UDP packets' rate in the case of the video resolution 720 p.

Online video streaming predominantly produces intermittent spikes in Ethernet frames tagged as UDP traffic. The frequency of UDP data transfer was influenced by the video resolution, with higher resolutions resulting in increased data traffic. A cross-correlation analysis was performed between the data from the packet sniffer and Wireshark. The results indicated correlation coefficients of 0.78 for the 1080 p video resolution, 0.92 for 720 p, and 0.57 for 480 p. These lower correlation coefficients were due to the intermittent nature of data transfers and the lack of synchronization between the packet sniffer and the PC running Wireshark. As a result, the sporadic data transfers exhibited variable delays when recorded via the packet sniffer and Wireshark, which affected the correlation between

these two datasets. We also measured the total UDP data transfer for both the packet sniffer and the PC running Wireshark, and the results are summarized in Table 1.

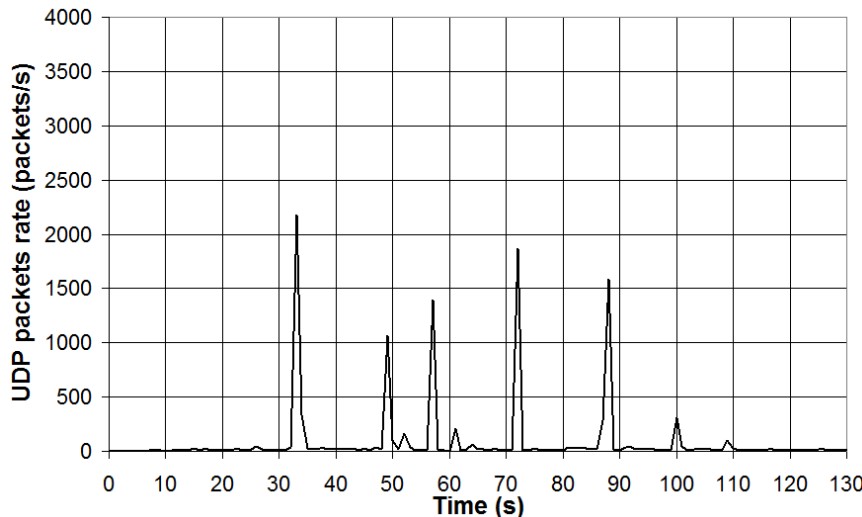

**Figure 25.** UDP packets' rate in the case of the video resolution 480 p.

**Table 1.** Total UDP data transferred in the case of a YouTube video with resolutions of 1080 p, 720 p, and 480 p for the packet sniffer and the PC running Wireshark.

| Device | Video Resolution | | |
|---|---|---|---|
| | 1080 p | 720 p | 480 p |
| Packet sniffer | 56.72 MB | 33.26 MB | 20.44 MB |
| PC running Wireshark | 56.51 MB | 33.14 MB | 20.27 MB |

As a third test, we conducted a file download (Arduino IDE, file size 197 MB) using the 'wget' command in Linux. This test employed the TCP protocol for data transfer, and the TCP data rate (data transferred per second) is illustrated in Figure 26.

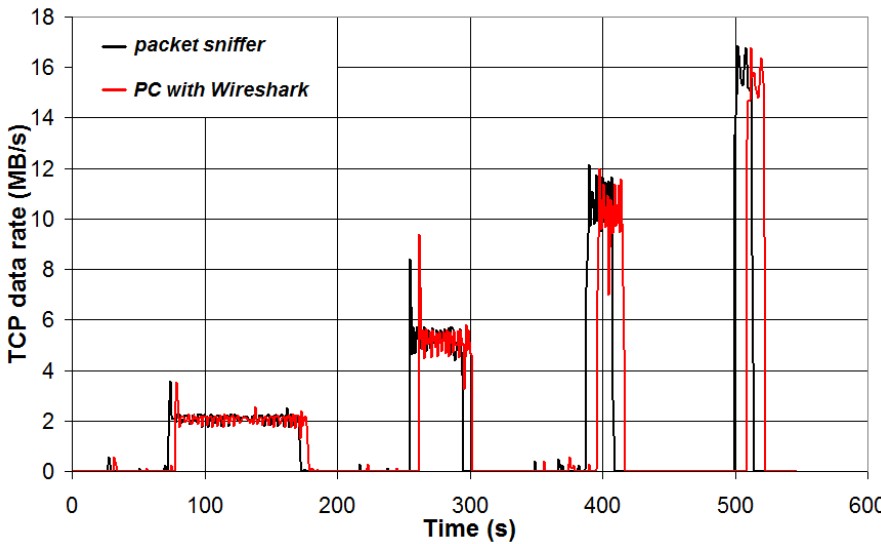

**Figure 26.** TCP data rate, in MB/s, in the case of file download using the Linux command "wget" for the packet sniffer and the PC running Wireshark.

The file was downloaded on four occasions, each time with different bandwidth limits: 2 MB/s, 5 MB/s, 10 MB/s, and 15 MB/s. A cross-correlation analysis was performed,

which revealed that the Wireshark data exhibited an approximate 9 s delay compared to the packet sniffer data. Despite this delay, the correlation coefficient remained notably high at 0.97.

Finally, we conducted a test to verify the packet sniffer's ability to accurately test for the rules defined in Section 4.2. Two distinct files were downloaded using the Linux 'wget' command, each from a different IP address (82.197.215.15 and 104.18.12.241). The rules were configured to solely permit data transfer from the IP address 104.18.12.241. The results of this test are depicted in Figure 27, where it is evident that the TCP data downloaded from 104.18.12.241 are correctly labeled as 'allowed data', while the data from 82.197.215.15 are flagged as 'data with rules violation'.

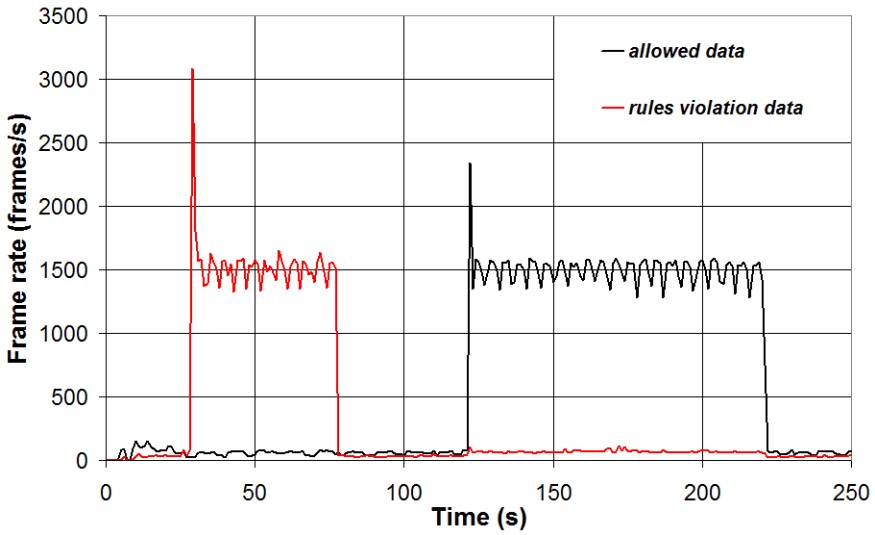

**Figure 27.** TCP packets' rate in the case of file download from two different IP addresses (104.18.12.241 and 82.197.215.15) using the Linux command "wget" for the packet sniffer.

## 6. Ethernet Data Analyses for Higher Data Rates

A hardware-based packet sniffer was designed for analyzing the Ethernet traffic at a maximum data rate of 1 Gbit/s. However, more recent Ethernet standards now allow for higher data speeds, such as 10 Gbit/s and 100 Gbit/s.

Preliminary efforts were made to extend the capabilities of the packet sniffer to operate at the higher data rate of 10 Gbit/s. For this purpose, we utilized the KC705 development board to create a packet generator running at 10 Gbit/s. The transmitted Ethernet frames were looped back on the receive channel of the same board. To accommodate this higher data rate, we switched from the RJ45 port to the small form-factor pluggable (SFP) port on the development board, enabling data transfer via optical fibers.

In this configuration, the physical layer of Ethernet transmission was managed using a PHY module instantiated within the FPGA, replacing the external PHY chip (Marvell M88E1111-BAB1C000), which can handle a maximum data speed of 1 Gbit/s. The IP module 10 G Ethernet Subsystem version 3.1, provided by Xilinx, was employed to handle layers one and two of the OSI protocol.

Figure 28 illustrates simulations of single-frame data transfers, specifically, a UDP packet with a length of 60 bytes. In the case of a 1 Gbit/s data rate, data were acquired with a 1-byte data width using a 125 MHz clock. For the 10 Gbit/s data rate, the data width was increased to 8 bytes with a 156.25 MHz clock. In summary, with a 1 Gbit/s rate, a 60-byte frame requires 60 clock cycles for data transfer, while at 10 Gbit/s, the number of clock cycles is reduced to eight. For the 100 Gbit/s data rate, data were acquired with a 64-byte data width, allowing for a 60-byte frame to be transferred in a single clock cycle.

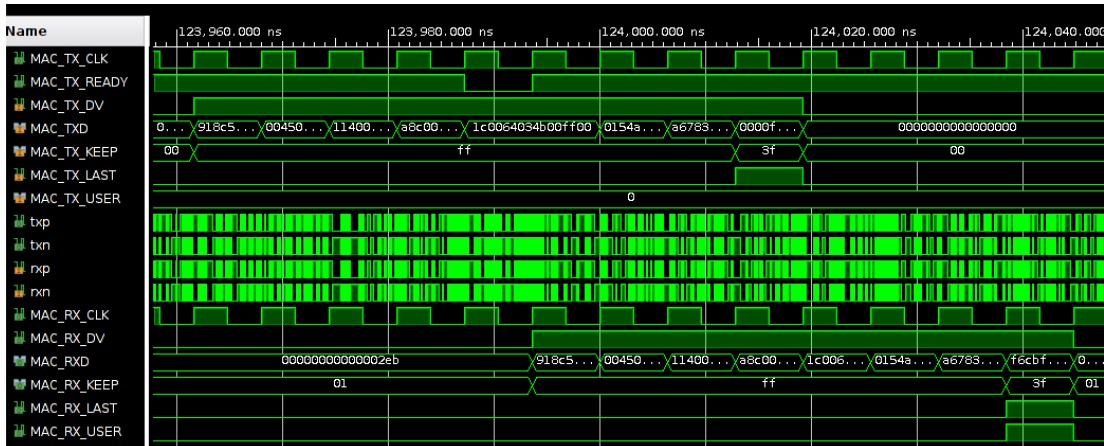

**Figure 28.** Signal waveforms in the case of data transmission and reception, with a 10 Gbit/s data rate.

The reduction in the number of clock cycles required for data frame transfers at higher data rates poses challenges for critical timing in frame analysis and packet sniffer rule verification. In the current version (1 Gbit/s), the timing is sufficient to verify up to 256 different packet sniffer rules. However, higher data rates demand stringent timing requirements, necessitating the redesign of the frame analysis hardware. In the case of a 10 Gbit/s data rate, we implemented 64 different packet sniffer rules that were checked in groups of 16 using a pipeline with a 4 clock cycle delay. Preliminary experimental tests, performed with a specially designed packet generator operating at 10 Gbit/s, confirmed the system's reliable and efficient performance.

## 7. Comparison with the State-of-the-Art

The performance of our proposed packet sniffer was compared to similar FPGA-based systems from the existing literature, and the results are summarized in Table 2. The system developed by Stój et al. utilizes the same FPGA device family as our work but lacks data traffic statistics collection and is limited to a 1 Gbit/s data rate [27]. Our preliminary results indicated that our system can be extended to operate at 10 Gbit/s. Pal's packet sniffer, 'E-Sniff', was implemented on a FPGA (Altera Cyclone® II 2C35 FPGA device on a DE2 development board) (Intel-Altera) and utilizes a soft processor and custom hardware to capture and display packets on a VGA monitor [28]. However, it supports lower data rates of 10 Mbit/s and 100 Mbit/s. Song et al.'s system was deployed on the Xilinx Virtex-E FPGA XCV2000E on the FPX platform, supporting a data rate for the OC48 network (2.5 Gbit/s). Nonetheless, it lacks features for generating statistics, and information regarding its supported protocols and PC interface is not available [29].

**Table 2.** Performance comparison of the proposed packet sniffer with similar systems implemented on FPGAs from the literature.

| Hardware Type | Data Rate (Gbit/s) | Supported Protocols | PC Interface | Statistics | Reference |
|---|---|---|---|---|---|
| Xilinx Kintex-7 | 1 | NA | PCI Express | No | [27] |
| Altera Cyclone II | 0.1 | About 15 | VGA monitor | No | [28] |
| Xilinx Virtex-E | 2.5 | NA | NA | No | [29] |
| Xilinx Virtex-4 | <1 | TCP, UDP, ICMP, and ARP | USB | Yes | [30] |
| Xilinx Virtex-7 + Linux firewall | 10 | TCP, UDP, ICMP, and ARP | PCI Express | No | [31] |
| Xilinx Virtex-4 | NA | TCP | NA | No | [32] |
| Xilinx Kintex-7 | 1–10 | TCP, UDP, ICMP, and ARP | UART | Yes | This work |

In 2009, Faria et al. introduced an Ethernet sniffer based on FPGA technology (Xilinx Virtex-4 XC4VFX140), with an interface to the host PC via an EZ-USB FX2 USB module from

Cypress (CY7C68013-100AC), which supports USB 2.0 high-speed connections. However, the supported data rate is below 1 Gbit/s [30]. HyPaFilter is a hybrid classification system comprising a FPGA development board (Xilinx VC709, with a Virtex-7 device) and a Linux software firewall on a PC. This system supports high data rates (10 Gbit/s) and utilizes a PCI Express interface but lacks data traffic statistics [31]. Ezzati et al.'s packet sniffer implemented packet filtering using neural networks and achieved an accuracy of higher than 97% [32]. It was implemented on a Xilinx Virtex-4 FPGA device (xc4vlx15), but a full working system was not developed, and information regarding the data rate is unavailable.

## 8. Conclusions

This article provided a comprehensive description of a stateless packet sniffer implemented on a FPGA. This packet sniffer was developed using a commercial FPGA development board (Xilinx's KC705), and is capable of supporting data transfer rates of up to 1 Gbit/s, with promising initial results indicating the potential for reliable data analysis, even at rates of 10 Gbit/s.

This system was engineered to meticulously analyze various types of Ethernet frames, including ARP, IP, UDP, TCP, and ICMP. It calculates essential frame attributes, such as MAC addresses, IP addresses, source ports, and destination ports, and assesses potential data threats based on user-defined rules.

Extensive testing was conducted under controlled conditions, which involved generating sets of Ethernet frames using a custom packet generator. Additionally, this system was evaluated under real-world internet traffic conditions. The performance tests covered certain scenarios, like video streaming and simulating an ICMP attack, demonstrating the packet sniffer's capability to reliably detect potential threats within the received data.

Future research endeavors in this domain will aim to enhance the packet sniffer's speed, with further testing to accommodate a 10 Gbit/s data rate, as well as upgrading the hardware to support 100 Gbit/s data rates.

Applications for this technology range from network monitoring to securing high-speed vertical internet connections. FPGAs have emerged as the optimal choice for these applications, primarily due to their inherent attributes of low latency, high throughput, and straightforward system reconfigurability.

**Author Contributions:** Conceptualization, M.G., A.G. and F.A.; methodology, M.G., A.G. and F.A.; software, M.G. and F.A.; validation, M.G. and F.A.; formal analysis, M.G. and F.A.; investigation, M.G. and F.A.; resources, M.G. and F.A.; data curation, M.G.; writing—original draft preparation, M.G.; writing—review and editing, M.G., A.G., F.A. and M.P.; visualization, M.G., A.G. and F.A.; supervision, A.G. and M.P.; project administration, A.G. and M.P.; funding acquisition, A.G. All authors have read and agreed to the published version of the manuscript.

**Funding:** The Italian Ministry of University and Research, Grant/Award Number: J45F21002000001; "Alma Idea 2022" Linea di Intervento A (D.M. 737/2021); the Italian Ministry of Industry Incentives (MISE); and the Ministry of University and Research (MUR). In addition, this work was partially supported by project SERICS (PE00000014) under the MUR National Recovery and Resilience Plan funded by the European Union—NextGenerationEU.

**Data Availability Statement:** Data sharing not applicable. No new data were created or analyzed in this study. Data sharing is not applicable to this article.

**Acknowledgments:** The authors would like to thank the National Institute for Nuclear Physics (INFN, Bologna division) and the National Center for Frame Analysis (CNAF, Bologna division) for their support in the development and testing of the presented packet sniffer.

**Conflicts of Interest:** The authors declare no conflict of interest.

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
