# Peer review of "A Highly Configurable Packet Sniffer Based on Field-Programmable Gate Arrays for Network Security Applications"

_electronics, doi:10.3390/electronics12214412_

Round 1
Reviewer 1 Report
This paper proposed a packet sniffing method based on FPGA. However, the novel contribution of the proposed method can not be found. Besides, some SOTA methods like ASIC-based or other FPGA-based SHOULD be compared to verify the proposed one.
N/A
Author Response
Response to reviewers’ comments
The authors appreciate the Reviewer comments, useful to improve the paper quality. The revised manuscript has been modified according to Reviewer suggestions and a step by step answer for all comments is provided. Modifications to the original manuscript are highlighted in green.
Reviewer #1
This paper proposed a packet sniffing method based on FPGA. However, the novel contribution of the proposed method can not be found. Besides, some SOTA methods like ASIC-based or other FPGA-based SHOULD be compared to verify the proposed one.
In the revised paper it is clearly pointed out in the introduction the advantages and disadvantages of designing a network security system with ASICs or FPGAs as well as the field of application of the proposed system.
In lines 71-77: “It is common for commercial firewall and packet sniffer products to be based on Application Specific Integrated Circuits (ASICs), which are highly optimized devices designed and manufactured for a specific application. They offer top performance in terms of speed, power consumption, and production cost per unit. However, ASICs are also characterized by complex design and high non-recurrent design costs, which make them ideal for high production volumes. Field Programmable Gate Arrays (FPGAs) are more suitable for products designed for small production volumes and research projects. These semiconductor devices feature quick design steps and negligible non-recurrent engineering costs.”
In lines 104-110: “In this work, an Ethernet packet sniffer based on FPGA is proposed. The system is intended for applications in data protection for universities and research institutes. As discussed by Ulven and Wangen in 2021, data breaches and cyberattacks represent a severe problem in higher education institutions and universities [33]. Since we estimate a low number of produced devices, the system was designed on FPGA which allows quick development and negligible non-recurrent engineering costs. Moreover, the system has been designed for easy reconfigurability, where the rules to discriminate between safe and potentially dangerous data can be decided by the user and uploaded by serial communication using a PC”.
Moreover, a new section (Section 7) has been added where the performance of the proposed system is compared with similar systems from literature.
The English language was thoroughly proofread, resulting in the correction of numerous errors and typos.
Reviewer 2 Report
The paper explore an area of interest to researcher, yet some of the comments that need to be carried.
Please add some related work, show the motivation, althought it waa mentioned but it was not clear . You can do that in the introduction.
The refrences are short and can be extend to include more recent work. Please explain and elaboraye more on figures, 7 and 9.
The expermintal part that was conducted, was there any challanges or issues?
Please indicate any negative feedback or any unwanted result that you encounter while conducting the experiment.
Some minor english editing is required
Author Response
Response to reviewers’ comments
The authors appreciate the Reviewer comments, useful to improve the paper quality. The revised manuscript has been modified according to Reviewer suggestions and a step by step answer for all comments is provided. Modifications to the original manuscript are highlighted in green.
Reviewer #2
The paper explores an area of interest to researcher, yet some of the comments that need to be carried.
Please add some related work, show the motivation, although it was mentioned but it was not clear. You can do that in the introduction.
The motivation of the work is now discussed at the end of the introduction “In this work, an Ethernet packet sniffer based on FPGA is proposed. The system is intended for applications in data protection for universities and research institutes. As discussed by Ulven and Wangen in 2021, data breaches and cyberattacks represent a severe problem in higher education institutions and universities [33]. Since we estimate a low number of produced devices, the system was designed on FPGA which allows quick development and negligible non-recurrent engineering costs. Moreover, the system has been designed for easy reconfigurability, where the rules to discriminate between safe and potentially dangerous data can be decided by the user and uploaded by serial communication using a PC”.
The references are short and can be extended to include more recent work.
In the revised paper 12 new references have been added.
Please explain and elaborate more on figures, 7 and 9.
In the revised paper more discussion is added on figures 7 and 9.
The experimental part that was conducted, was there any challenges or issues? Please indicate any negative feedback or any unwanted result that you encounter while conducting the experiment.
In the revised paper two issues encountered during the tests on the developed system have been reported. In lines 177-182 “To ensure the packet sniffer's accuracy, using a packet generator instead of a PC with software to generate Ethernet frames is important, which may produce uncertain results due to uncontrolled data traffic (a PC with a network card generates control frames in addition to the target Ethernet frames). The experimental setup includes a packet generator that produces Ethernet frames with selectable variable data rates, and a packet sniffer that analyses Ethernet traffic and generates data statistics based on frame fields”. In lines 365-367 “To accommodate this experimental setup, we implemented the optional virtual local area network (VLAN) field in the Ethernet frame within the packet sniffer. This adjustment was necessary because the traffic mirrored to port 10 carries the VLAN tag”.
The English language was thoroughly proofread, resulting in the correction of numerous errors and typos.
Reviewer 3 Report
The following aspects should be addressed, before any further processing of this paper may take place.
1. The experimental setup that is presented is not clearly indicative of this approachțs relevance for real-world use cases. Thus, the authors should suggest, in a clearer manner, the real-world usefulness and relevance of their proposed model. This should be justified with consistent conceptual remarks.
2. The authors should comparatively assess their work against similar relevant approaches. Thus, a separate sub-section should be addedd, which compares the reported model against 4-5 other relevant solutions.
3. The list of references may be enriched with 10-15 up to date papers.
4. The English language should be improved through, at least, one round of proofreading.
The English language should be improved through, at least, one round of proofreading.
Author Response
Response to reviewers’ comments
The authors appreciate the Reviewer comments, useful to improve the paper quality. The revised manuscript has been modified according to Reviewer suggestions and a step by step answer for all comments is provided. Modifications to the original manuscript are highlighted in green.
Reviewer #3
The following aspects should be addressed, before any further processing of this paper may take place.
The experimental setup that is presented is not clearly indicative of this approach relevance for real-world use cases. Thus, the authors should suggest, in a clearer manner, the real-world usefulness and relevance of their proposed model. This should be justified with consistent conceptual remarks.
In the revised paper the real-world use case of the proposed packet sniffer is clearly presented at the end of the introduction: “In this work, an Ethernet packet sniffer based on FPGA is proposed. The system is intended for applications in data protection for universities and research institutes. As discussed by Ulven and Wangen in 2021, data breaches and cyberattacks represent a severe problem in higher education institutions and universities [33]. Since we estimate a low number of produced devices, the system was designed on FPGA which allows quick development and negligible non-recurrent engineering costs. Moreover, the system has been designed for easy reconfigurability, where the rules to discriminate between safe and potentially dangerous data can be decided by the user and uploaded by serial communication using a PC”.
The authors should comparatively assess their work against similar relevant approaches. Thus, a separate sub-section should be added, which compares the reported model against 4-5 other relevant solutions.
As suggested by the Reviewer, a new section (Section 7) has been added where the proposed packet sniffer is compared with other similar systems on FPGA from the literature.
The list of references may be enriched with 10-15 up to date papers.
In the revised paper, 12 new references have been added.
The English language should be improved through, at least, one round of proofreading.
The English language was thoroughly proofread, resulting in the correction of numerous errors and typos.
Reviewer 4 Report
In this paper, the authors focused on the growth of web applications and online transactions because they have led to an increase in cyberattacks, which pose a significant threat to digital services. The authors noticed firewalls are deployed on computing nodes and network devices to detect and block malicious packets. Firewalls can be software-based or hardware-designed, with software-based solutions offering flexibility and easy upgradability. Also, they noticed hardware implementation is the only viable solution for high data rates. This paper presented a packet sniffer designed on FPGA with a 1 Gbit/s data transfer rate, implemented on Xilinx's KC705 development board.
This paper considers significant security issues. We widely use the Internet for many activities, sometimes without critical thinking about our clicking and appropriate knowledge about security.
The paper is well-organised. The literature review is appropriate.
Two remarks:
- The authors should highlight their contributions in the Introduction.
- The authors should separate the article structure in the next paragraph. Now, the article structure is a part of an earlier paragraph.
Author Response
Response to reviewers’ comments
The authors appreciate the Reviewer comments, useful to improve the paper quality. The revised manuscript has been modified according to Reviewer suggestions and a step by step answer for all comments is provided. Modifications to the original manuscript are highlighted in green.
Reviewer #4
In this paper, the authors focused on the growth of web applications and online transactions because they have led to an increase in cyberattacks, which pose a significant threat to digital services. The authors noticed firewalls are deployed on computing nodes and network devices to detect and block malicious packets. Firewalls can be software-based or hardware-designed, with software-based solutions offering flexibility and easy upgradability. Also, they noticed hardware implementation is the only viable solution for high data rates. This paper presented a packet sniffer designed on FPGA with a 1 Gbit/s data transfer rate, implemented on Xilinx's KC705 development board.
This paper considers significant security issues. We widely use the Internet for many activities, sometimes without critical thinking about our clicking and appropriate knowledge about security.
The paper is well-organised. The literature review is appropriate.
Two remarks:
The authors should highlight their contributions in the Introduction.
In the revised paper, at the end of the introduction, the contribution of our work is discussed: “In this work, an Ethernet packet sniffer based on FPGA is proposed. The system is intended for applications in data protection for universities and research institutes. As discussed by Ulven and Wangen in 2021, data breaches and cyberattacks represent a severe problem in higher education institutions and universities [33]. Since we estimate a low number of produced devices, the system was designed on FPGA which allows quick development and negligible non-recurrent engineering costs. Moreover, the system has been designed for easy reconfigurability, where the rules to discriminate between safe and potentially dangerous data can be decided by the user and uploaded by serial communication using a PC”.
The authors should separate the article structure in the next paragraph. Now, the article structure is a part of an earlier paragraph.
As suggested by the Reviewer, the structure of the article is now presented in a separate section (Section 2).
The English language was thoroughly proofread, resulting in the correction of numerous errors and typos.
Round 2
Reviewer 1 Report
I suggest that this version can be accepted.
I suggest that this version can be accepted.
Reviewer 3 Report
ALthough section 7 appears rather undersized, I appreciate that the paper may be considered, in its revised version, for further processing. Additional English proofreading is recommended during the editorial processing stage.
Additional English proofreading is recommended during the editorial processing stage.